# Evaluation of Elastic Properties and Conductivity of Chitosan Acetate Films in Ammonia and Water Vapors Using Acoustic Resonators [note 1]

**DOI:** 10.3390/s20082236

**Published:** 2020-04-15

**Authors:** Boris D. Zaitsev, Andrey A. Teplykh, Fedor S. Fedorov, Artem K. Grebenko, Albert G. Nasibulin, Alexander P. Semyonov, Irina A. Borodina

**Affiliations:** 1Kotelnikov Institute of Radio Engineering and Electronics, Russian Academy of Sciences, Saratov Branch, 410019 Saratov, Russia; teplykhaa@mail.ru (A.A.T.); alex-sheih@yandex.ru (A.P.S.); borodinaia@yandex.ru (I.A.B.); 2Skolkovo Institute of Science and Technology, 3 Nobel St., 121205 Moscow, Russia; F.Fedorov@skoltech.ru (F.S.F.); A.Grebenko@skoltech.ru (A.K.G.); A.Nasibulin@skoltech.ru (A.G.N.); 3Moscow Institute of Physics and Technology, Institute Lane 9, 141701 Dolgoprudniy, Russia; 4Department of Chemistry and Materials Science, Aalto University, 00076 Espoo, Finland

**Keywords:** piezoelectric resonators, resonant frequency, electrical impedance, chitosan acetate films, electrical conductivity, concentration of ammonia, air humidity

## Abstract

Novel bio-materials, like chitosan and its derivatives, appeal to finding a new niche in room temperature gas sensors, demonstrating not only a chemoresistive response, but also changes in mechanical impedance due to vapor adsorption. We determined the coefficients of elasticity and viscosity of chitosan acetate films in air, ammonia, and water vapors by acoustic spectroscopy. The measurements were carried out while using a resonator with a longitudinal electric field at the different concentrations of ammonia (100–1600 ppm) and air humidity (20–60%). It was established that, in the presence of ammonia, the longitudinal and shear elastic modules significantly decreased, whereas, in water vapor, they changed slightly. At that, the viscosity of the films increased greatly upon exposure to both vapors. We found that the film’s conductivity increased by two and one orders of magnitude, respectively, in ammonia and water vapors. The effect of analyzed vapors on the resonance properties of a piezoelectric resonator with a lateral electric field that was loaded by a chitosan film on its free side was also experimentally studied. In these vapors, the parallel resonance frequency and maximum value of the real part of the electrical impedance decreased, especially in ammonia. The results of a theoretical analysis of the resonance properties of such a sensor in the presence of vapors turned out to be in a good agreement with the experimental data. It has been also found that with a growth in the concentration of the studied vapors, a decrease in the elastic constants, and an increase in the viscosity factor and conductivity lead to reducing the parallel resonance frequency and the maximum value of the real part of the electric impedance of the piezoelectric resonator with a lateral electric field that was loaded with a chitosan film. This leads to an increase in the sensitivity of such a sensor during exposure to these gas vapors.

## 1. Introduction

Advances in gas sensors are primarily associated with an improvement of their characteristics (sensitivity, selectivity, stability, fast response/recovery time, power consumption) based on the evaluation of the sensing mechanism. Knowledge of the mechanism enables the rational design of materials that are complementary to the modification of sensor architecture to achieve the best performance. The operation of gas sensors is defined by both a sensing material in which properties are changed upon interaction with gas vapors under study and a physical transducer that transforms these changes to a signal that is monitored by a human [1,2,3]. Recent attention to electromechanical resonant transducers, like piezoelectric resonators with a lateral electric field, is stimulated by a broad scope of potential applications, i.e. including various sensors [4,5,6,7,8]. These resonators are sensitive to changes in the electrical conductivity and mechanical impedance of the medium in contact with the free side of the resonator, whereas the electrodes are located on the other side of the piezoelectric plate. In particular, the resonators with a lateral electric field have been applied as sensors to assess the viscosity and conductivity of a contacting liquid [9,10]. The effect of the conductivity of a thin layer located near the resonator on its characteristics has also been evaluated [11,12]. It is shown that a change in the conductivity of the layer significantly influences the frequency of parallel resonance and the maximum value of the real part of the electrical impedance. Thus, while assuming that the adsorption of gas molecules changes the conductivity of the sensitive film, one can employ the resonator that is covered by such film as a gas sensor.

Primarily candidates to serve a sensitive layer are films of chitosan, a biopolymer that consists of ß-1,4 linked GlCN and GlcNAc units with hydroxyl-, amino-, and some number of N-acetyl functional groups [13,14]. It can be employed in the form of various derivatives or composites and it is often represented by a gel structure [14]. Applications of these materials include primarily electrochemical biosensors [15,16]. Still, these materials are also applied in both chemoresistive [17,18] and mass [19] sensors at room temperature, owing to the sensing mechanism that should include swelling due to vapor adsorption and dipole-dipole interaction/hydrogen bonding influenced by analyte nature and diffusion regime [20]. The film response depends on the polar component of solubility, dielectric permittivity, and the size of the molecules of vapors [20]. Recently, we have shown that the resonance properties of the structure “resonator with a lateral electric field—air gap—chitosan film” [21] strongly change in the presence of vapors of volatile compounds, such as water, ethyl alcohol, and ammonia. We have revealed that, in the presence of vapor of volatile liquids, the conductivity of such films changes by 1–2 orders of magnitude, however, the measured response and relaxation times of the conductivity of the films have turned out to be significantly less than the response and relaxation times of the resonance frequency and the maximum value of the real part of the electrical impedance of the structure. This fact points that the mechanical properties of the films are also changed by vapor exposure; these changes are more inertial. Still, the mechanism of the sensing due to the appearance of polar vapors needs to be further addressed. In this case, the use of physical transducer, such as the resonator with a lateral electric field covered by a chitosan film, represents the most advantageous design.

We have selected ammonia and water vapor for these tests. Recent studies suggest that conductive polymers, nanocomposites, and graphene films are rather promising for ammonia sensing [22,23,24,25]. The earlier study [21] of the sensitivity of chitosan films to ammonia, as well as to water vapor, has also raised a number of questions that will be partially resolved in this article.

This article is devoted to the study of changes in the mechanical properties and conductivity of a chitosan acetate film in the presence of volatile liquid vapors and to the research of the parameters of a sensor that is based on a lateral electric field excited resonator with chitosan film.

## 2. Materials and Methods

### 2.1. Preparation of Chitosan Acetate Films

In this study, we applied chitosan acetate, because it is one of the most investigated and widely available water-soluble derivative. We plan to measure the change in the mechanical parameters of chitosan films (longitudinal and shear elastic constants and viscosity factor) in the presence of vapor of volatile liquids. Therefore, the chitosan film, which was deposited on the surface of piezoelectric resonator, should have the plane—parallel sides and smooth surface. We were not able to make the necessary films of other chitosan derivatives, like chitosan lactate and glycolate of several tens of microns thick, but these parameters were met with chitosan acetate. It also has the easiest solution preparation protocol among the organic acid salts of chitosan (that provide a smoother surface, when compared to inorganic acid salts of chitosan). Chitosan acetate was produced by heterogeneous synthesis, i.e., briefly, chitosan (Bioprogress LCC, Moscow, Russia) with molecular weight, 150–200 kDa, was added to a solution of acetic acid (Sigma Aldrich) in the water-ethanol mixture. The mixture was stirred for 3 h at 50 °C. The obtained precipitate was filtered and then dried at a rotary evaporator while using residual pressure 15 mbar and temperature of 50 °C. We have prepared a 1.5% aqueous solution using the obtained chitosan acetate. This solution was drop-casted (ca. 0.5 mL) at the electrodes of the piezoceramic resonator or on the glass surface and was then dried in ambient conditions for 24 h. The film thickness was measured to vary in the range from 20 to 30 μm. The drop-casting procedure, followed by drying, was performed several times in the case that greater thickness was required.

### 2.2. Topology and Profile of Chitosan Acetate Films

Atomic Force Microscopy (AFM) imaging was performed with a Bruker Multimode V8 device operating in the PeakForce tapping regime with HA_CNC cantilevers (k = 1.5 N/m, Optek, Russia).

### 2.3. Determination of the Elastic and Viscosity Coefficients of Chitosan Films

The elastic and viscosity coefficients of the chitosan films were determined by the method of acoustic spectroscopy while using a piezoelectric resonator with a longitudinal electric field [14]. The resonator represented a disk 2 mm thick and 22 mm in diameter made of piezoceramics (Ba_0.24_Pb_0.75_Sr_0.01_(Ti_0.47_Zr_0.53_)O_3_) (Figure 1a). The polar axis was oriented perpendicular to the disk. Both sides of the resonator were completely coated with silver electrodes 5 μm thick. Initially, the frequency dependence of the electrical impedance module of the resonator without the film was measured in the frequency range of 50–1450 kHz using the impedance analyzer E4990A (Keysight Technologies). This dependence represented a sequence of resonance peaks, each corresponding to a specific oscillation mode.

The frequency dependence of the electric impedance module was also calculated by the finite element method while using the values of elastic constants, viscosity coefficients, piezoelectric constants, and permittivity of ceramics found in the literature. In this case, the density of piezoceramics was determined previously based on the mass and the volume of the resonator. Subsequently, by fitting the indicated theoretical dependencies to the experimental ones, the refined material constants of piezoceramics were determined while using the Nelder–Mead algorithm [26,27]. Afterward, the studied film was deposited on one side of the resonator directly onto the electrode, and the measurement of the frequency dependence of the electric impedance module was repeated. Further, this dependence was calculated by the finite element method and two elastic constants (c_11_ and c_44_) and the viscosity factor (η) of the film were determined while using the Nelder–Mead algorithm. The density of the film was determined from the measured mass and volume.

In this work, the mechanical properties of a chitosan acetate film were determined in the presence of vapors of volatile liquids, such as ammonia and water. For this purpose, a special sealed chamber was fabricated, where a resonator with a film and a container with volatile liquid were placed (Figure 1b). Initially, the measurements were carried out in air. Subsequently, a container with liquid ammonia or water was put in the chamber, the cover was tightly closed, and the measurements were carried out in the presence of ammonia or water vapor for 2–3 h. After that, the cover was opened, and the measurements were continued in air. The frequency dependences of the impedance module were recorded with an interval of 45 s. Such a measurement cycle allowed for finding the time dependences of the longitudinal (c_11_) and shear (c_44_) elastic constants and viscosity factor (η) of the film. From the known values of the chamber volume (75 mL), the area of the open surface of the liquid container (2.8 cm^2^) and temperature (26 °C), the time dependencies of the concentration of ammonia and air humidity in the chamber were calculated. Afterwards, we constructed the dependencies of the elastic constants and viscosity factor of the chitosan film versus the concentration of ammonia and air humidity in the chamber.

### 2.4. Measurement of the Electrical Conductivity of the Film

For the study of electrical conductivity of the film, a sample was made on the basis of a glass plate with the shear dimensions of 16 × 27 mm^2^. Two aluminum electrodes were deposited on one side of the plate with a center gap of 10 mm wide. A film of chitosan acetate was deposited on the central part of the plate, so that the edges of the film touched the electrodes with an overlap of ca. 1 mm. The sample was placed in the aforementioned sealed chamber and the electrodes were connected through auxiliary wires to an impedance analyzer E4990A (Keysight Technologies), which operated in the “conductivity–capacitance” measurement mode at the frequency of 98 kHz. The specific conductivity of the film was determined using the known geometry of the chitosan film and the known total conductivity. The measurement cycle was exactly the same as described in Section 2.3 for determining the mechanical properties of the film: first, the conductivity was measured in air, then in the test vapor, and after that again in the air. Measurements were carried out with an interval of 45 s. As a result, we found the time dependence of the specific conductivity of the film in the air and the studied vapors. The dependencies of the conductivity of the chitosan film on the concentration of ammonia and air humidity in the chamber were constructed based on the known time dependences of the concentration of ammonia and air humidity in the chamber during the measurement.

It should be noted that all experiments were carried out at least twice and the reproducibility of the results was about ± 2%. The measurements were carried out at a temperature of 26–27 °C, a pressure of 1000 hPa, and a humidity of 20%. As for stability over time, the results of our previous work [21], in which the resonator and the chitosan film were separated by an air gap, remain stable within 5% for one year.

### 2.5. The study of the Structure “Resonator with Lateral Electric Field—Film of Chitosan”

The piezoelectric resonator with lateral electric field was made of the plate of piezoceramics (Pb_0.95_Sr_0.05_(Ti_0.47_Zr_0.53_)O_3_) with a thickness of ~3 mm and with shear dimensions of 20 × 18 mm^2^ (Figure 1c). Two aluminum electrodes 200 nm thick with 7 × 20 mm^2^ dimensions and a 4 mm gap between them were deposited on one side of the plate. The polar axis of the piezoelectric was oriented perpendicular to the gap between the electrodes. The resonator was connected to an impedance analyzer to measure the frequency dependences of the real and imaginary parts of the electrical impedance. Three parallel resonances were detected at 68, 98, and 260 kHz. We analyzed the resonance properties of the sensor only near the resonant frequency of 98 kHz.

A film of chitosan acetate 60 μm thick was deposited on the free side of the resonator. We measured the frequency dependence of the real part of the electrical impedance of this resonator that was placed in the sealed chamber in the presence of vapors of volatile liquids with an interval of 45 s. The measurement cycle was the same as described in the previous paragraphs: first in the air, then in the test vapor, and after that again in the air. Measurements were carried out until saturation of the real part of the electrical impedance. This saturation corresponded to the saturated vapor pressure of the volatile liquid. Subsequently, the container was opened and measurements were carried out in the air.

A theoretical analysis of the frequency dependencies of the real and imaginary parts of the electric impedance of a resonator with a lateral electric field and a chitosan film on its free side was carried out by the finite element method while using the known geometry of resonator, electrodes, and film, and the known values of the elastic constants, viscosity factor, density, dielectric constant, and conductivity of piezoceramics and films.

## 3. Results & Discussion

### 3.1. Topology Studies

The study of the film local topology revealed the characteristic roughness of the film surface in a dry state of approximately 1 nm (RMS value), as depicted in Figure 2.

### 3.2. Determination of the Time Dependences of Ammonia Concentration and Air Humidity in a Closed Chamber

From the known values of the chamber volume (75 mL), the area of the open surface of the liquid container (2.8 cm^2^) and temperature (26 °C), the time dependences of the concentration of ammonia and air humidity in the chamber were calculated [28]. Figure 3 presents these dependencies.

### 3.3. Determination of Elasticity and Viscosity Coefficients of Films in the Vapor of Volatile Liquids

The elasticity and viscosity coefficients of the films were determined by acoustic spectroscopy while using the resonator with a longitudinal electric field, as already noted. Figure 4a shows, as an example, the measured frequency dependences of the electrical impedance modulus of the resonator with a chitosan acetate film in ammonia after exposure of two hours over the entire frequency range of 50–1450 kHz. A slight difference in these curves is visible in a narrow frequency range of 1200–1300 kHz (Figure 4b), which allowed for determining the effect of ammonia on the coefficients of elasticity and viscosity of the film.

For the calculation, we have used the fitted material constants of piezoceramics (Ba_0.24_Pb_0.75_Sr_0.01_(Ti_0.47_Zr_0.53_)O_3_) and determined the film characteristics in air, which are shown in Table 1:

Here, c_ij_ is the elastic constant, e_ij_ is the piezoconstant, ε_ij_ is the relative dielectric constant, ρ is the density, η is the viscosity factor.

Figure 5a shows the time dependences of the elastic constants and viscosity factor of chitosan acetate film. At time t = 0, the chamber began to be filled out with ammonia. At time t = 145 min, the chamber cover was opened and the analyzed values began to tend to the initial state. It is seen that, in the presence of ammonia, the elastic constants decreased with time and tended to reach saturation, and the viscosity factor increased. One can also see that the relaxation time was slightly more than 35 min. The dependencies of the elastic constants and viscosity factor on the concentration of ammonia in the chamber were constructed using the graph presented in Figure 3a (Figure 5b). Figure 5b shows that the elastic constants c_11_ and c_44_ monotonously decrease by 31% and 35%, respectively, with an increase in the concentration of ammonia from 0 to 0.17%. In this case, the viscosity factor in this concentration range increases by 2.5 times.

Subsequently, the described resonator with a film was studied in water vapor with the same measurement cycle as in ammonia. Figure 6a shows the dependencies of the elastic constants and the viscosity factor on time. The graph presented in Figure 3b allowed to construct the dependencies of the elastic constants and the viscosity factor on the air humidity in the chamber. Figure 6b depicts these dependencies. One can see that elastic modules did not practically change at increasing the humidity. At that the factor of viscosity increased by 1.4 times.

### 3.4. Measurement of the Conductivity of Films in Vapor of Volatile Liquids

Following the procedure that is described in Section 2, the specific conductivity of the chitosan acetate film was measured in air, in ammonia, and in water vapor. Figure 7a,b show the time dependencies of the specific conductivity of the chitosan film for ammonia and water vapor. It can be seen that with increasing the time the film conductivity in both cases increases. Figure 8a,b depict the dependencies of the film conductivity on the concentration of ammonia and air humidity in the chamber, respectively. The rise of the concentration of ammonia in the range of 0–0.17% leads to an increase in the conductivity by a factor of 110. The growth of the humidity from 20 up to 60% increases the conductivity by a factor of 7. One can see that ammonia has a stronger effect on the conductivity of the film.

The relaxation time in both cases equaled to 3 min.

### 3.5. Measurement of the Resonance Properties of the Structure “Resonator with a Lateral Electric Field-Chitosan Film” in the Vapor of Volatile Liquids

Figure 9a shows the experimental frequency dependence of the real part of the electric impedance of a resonator with a lateral electric field with a film of chitosan acetate in the air (curve 1), in water vapor with the humidity 45% (curve 2), and in ammonia with the concentration of 1600 ppm (curve 3) near the resonant frequency 98 kHz. One can see that, in the indicated vapors, the parallel resonance frequency and the maximum value of the real part of the electrical impedance decrease. The relative changes in the real part of the electrical impedance for water vapor and ammonia turned out to be 3.7 and 37%, respectively.

For the calculation, we have used the material constants of piezoceramics (Pb_0.95_Sr_0.05_(Ti_0.47_Zr_0.53_)O_3_) and film characteristics in ammonia and water vapor, which are shown in Table 2.

Here, c_ij_ is the elastic constant, e_ij_ is the piezoconstant, ε_ij_ is the relative dielectric constant, ρ is the density, σ is the specific conductivity.

This difference is associated with a more significant change in the conductivity and viscosity of the film in ammonia when compared with water vapor (Figure 5, Figure 6, and Figure 8). One can also see that the difference between curves 1 and 4 is insignificant, i.e. a film in air has practically no effect on the resonator. Figure 9b presents the theoretical frequency dependences, which turned out to be in good agreement with the experiment. It was assumed that all of the material constants for the material of piezoceramics did not depend on the presence of vapor of the studied liquids. As for the chitosan film, the elasticity and viscosity coefficients, as well as the electrical conductivity, were determined both in air and in the vapors of the analyzed liquids. Obviously, the high sensitivity of the resonance properties of the sensor in the vapor of the studied liquids is associated with a strong change in the conductivity and viscosity of the film. Thus, the presented effects, i.e. changes in the conductivity and mechanical properties, have a cumulative contribution to a change of the parallel resonance frequency and maximum value of the real part of the electrical impedance of a piezoelectric resonator with a lateral electric field being loaded by a chitosan film.

### 3.6. Discussion

The study showed the possibility of using a chitosan film to detect ammonia and estimate humidity when employed as a sensing layer in combination with piezoelectric resonators. At that, the sensitivity of the sensor to ammonia turned out to be significantly higher when compared to water vapor. This is because the ammonia significantly changes film parameters, such as longitudinal and shear elastic constants, viscosity factor, and conductivity. Water vapor increases the viscosity and conductivity to a lesser extent, but practically do not change the elastic constants. Water and ammonia vapors both support the change of hydroxyl ion conductivity in chitosan films, i.e. water can protonate free amino groups in the chitosan backbone also forming some hydroxyl ions (−NH_2_ + H_2_O → −NH_3_^+^ + OH^−^). This results in the growth of the charge carrier concentration and their mobility, which is supported by the mechanism of ionic transfer under hydration due to the movement of OH- ions in the electrical field created by NH_3_^+^ [29]. This yields conductivity change, while the adsorption of molecules favors the change in the mechanical properties.

Table 3 and Table 4, respectively, present the data on ammonia and humidity sensors obtained in current work and by other authors.

In Table 3, PPy means polypyrrole, TANIPANI means tannisulfonic acid doped polyaniline, and LOD is the limit of detection.

In Table 4, RGO-PVR film on THS means reduced graphene oxide-polyvinylpyrrolidone film on triboelectric humidity sensor.

A comparison of the obtained results with the data of other authors allows for us to draw the following conclusion.

Our results show that sensors that are based on a film of chitosan acetate can detect the presence of ammonia at a concentration of 100–1600 ppm. This is not a record, because modern materials generally cover this range, e.g., the sensitivity of the sensor coefficient of the chitosan film (~0.0062%/ppm) is lower in comparison with the sensor on TANIPANI/TiO_2_ (5.8%/ppm) [22]. Another advantage of such sensors is their short response and relaxation times. However, these sensors with record parameters can operate in an extremely narrow range of concentrations and are characterized by complex manufacturing technology. Moreover, these sensors, which are based on changing the resistance of the films in the presence of ammonia, are also very sensitive to humidity. Obviously, the readings of these ammonia sensors will also depend on the presence of other gases in the test volume. Since our sensor is less responsive to water vapor, it can be assumed that as an ammonia sensor it is less susceptible to humidity in the range of 20–60%. For example, the influence of humidity on the parameters of the sensor based on chitosan film is significantly weaker when compared to the humidity sensor that is based on RGO-PVR film on THS.

Another advantage of the sensor that is based on a piezoelectric resonator with a lateral electric field in comparison with resistive sensors is the ability to safely analyze combustible and explosive gases. This is because the resonator electrodes that are supplied with an electric voltage can easily be isolated from contact with the gas. Such a sensor might be useful, for example, in detecting an excessive amount of aviation kerosene vapor in jet aircraft containers. Other things being equal, such a sensor will never give a spark in such a dangerous container.

## 4. Conclusions

Thus, we have confirmed the earlier assumption that not only the conductivity of chitosan films changes in the vapor of volatile liquids, but also their mechanical properties. It has been shown that such analyte as ammonia yield changes in the electrical conductivity by an order of magnitude greater when compared to water. The viscosity of the film varies greatly both in water vapors and in ammonia. However, in ammonia, the longitudinal and shear elastic constants decrease significantly and, in water vapor, their changes are insignificant. In general, the results obtained show the possibility to develop a multi-parameter gas sensor based on resonators with a lateral and longitudinal electric field with a chitosan film applied. We have shown that there exist six parameters whose values are changed in the presence of gas. They are four film parameters: longitudinal and shear elastic constants, viscosity factor, and conductivity and two resonator parameters: parallel resonance frequency and the value of the real part of the electrical impedance at this frequency. Thus, it opens the possibility of developing a sensor (analyzer) in which the test gas is characterized by a point in six-dimensional space. Studies have shown that water vapor and ammonia can be confidently separated by such a combination of parameters. In the future, we will improve research methods and use other gases. Accordingly, the obtained results open up the possibility of developing a selective gas sensor that is based on a resonator with a lateral electric field and a chitosan film.

## Figures and Tables

**Figure 1 sensors-20-02236-f001:**
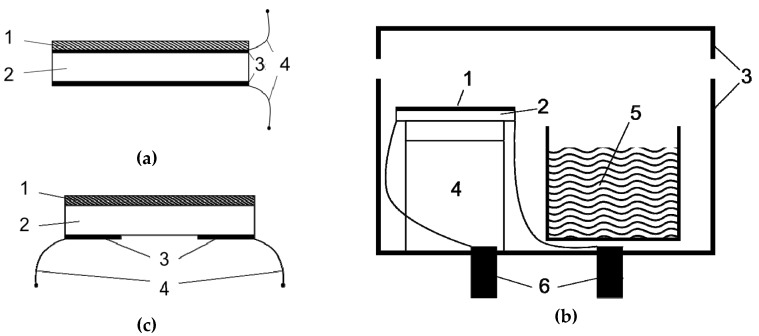
Schematic view of the experimental setup. Side views of disc resonator with a longitudinal electric field (**a**) and resonator with a lateral electric field (**c**): 1—film of chitosan acetate, 2—piezoceramics, 3—metal electrodes, 4—contacts. Gas chamber (**b**): 1—chitosan acetate film, 2—resonator, 3—camera body and sealing cover, 4—solid support with top low impedance support, 5—volatile liquid in the container, 6—electrodes for the impedance analyzer port.

**Figure 2 sensors-20-02236-f002:**
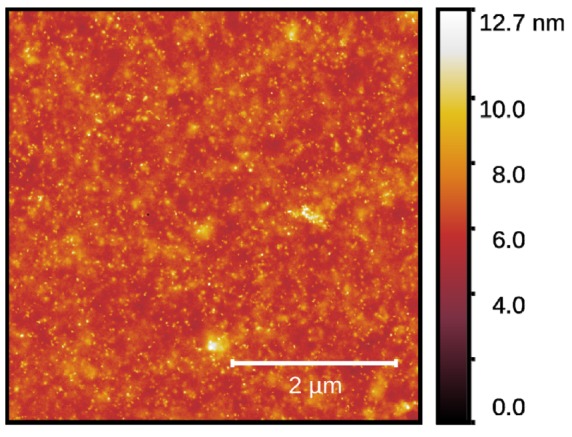
The topology of chitosan acetate. An Atomic Force Microscopy (AFM) 5 × 5 μm^2^ topology image of chitosan acetate film collected in the PeakForce tapping regime.

**Figure 3 sensors-20-02236-f003:**
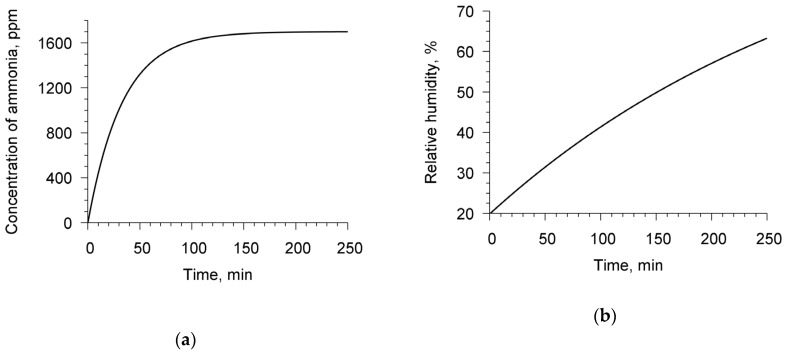
Computed time dependencies of ammonia concentration in air, by weight (**a**) and relative air humidity (**b**) in the gas chamber.

**Figure 4 sensors-20-02236-f004:**
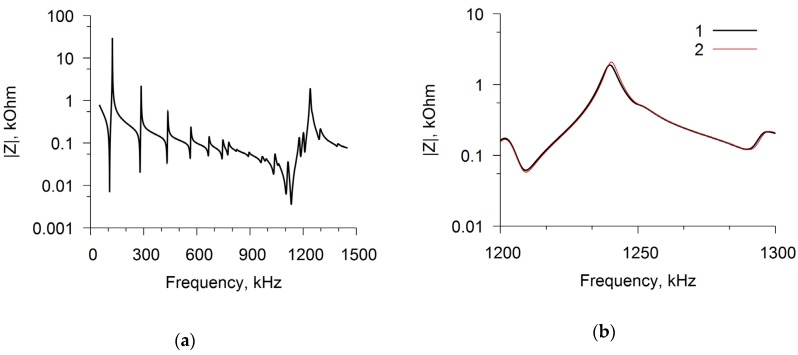
Frequency dependences of the electrical impedance module for a disk resonator with a film of chitosan acetate in the air (curve 1) and in ammonia with concentration 1600 ppm (curve 2). The frequency range is equal to 50–1450 kHz (**a**) and to 1200–1300 kHz (**b**).

**Figure 5 sensors-20-02236-f005:**
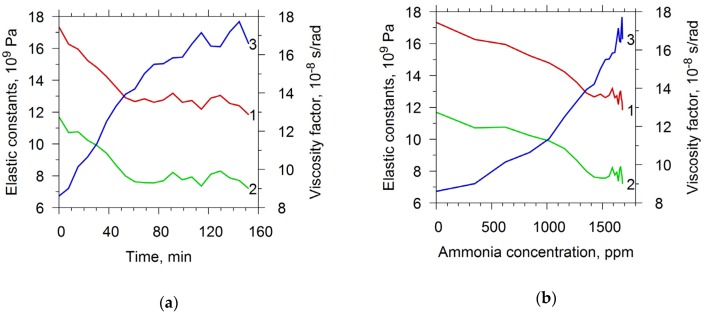
The material constants: c_11_ (curve 1), c_44_ (curve 2) and viscosity factor η (curve 3) of chitosan acetate film as functions of the time (**a**) and of the ammonia concentration (**b**).

**Figure 6 sensors-20-02236-f006:**
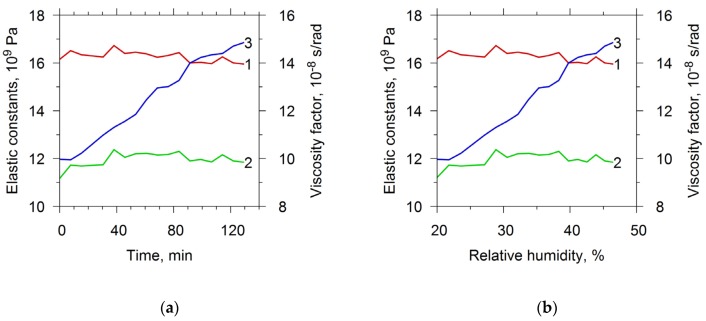
The material constants: c_11_ (curve 1), c_44_ (curve 2) and viscosity factor η (curve 3) of chitosan acetate film as function of the time (**a**) and of the relative humidity (**b**).

**Figure 7 sensors-20-02236-f007:**
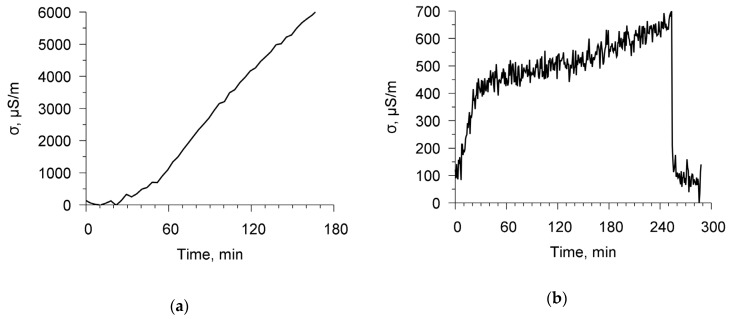
The time dependencies of specific conductivity of chitosan acetate film in ammonia (**a**) and water vapor (**b**) at frequency 98 kHz.

**Figure 8 sensors-20-02236-f008:**
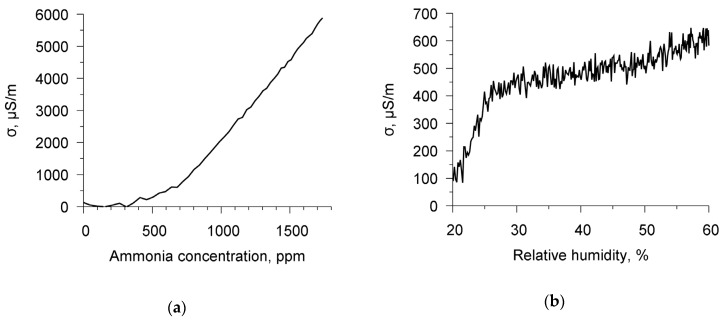
Dependencies of specific conductivity of chitosan acetate film on the concentration of ammonia in the air (**a**) and on the relative humidity of air (**b**) at frequency 98 kHz.

**Figure 9 sensors-20-02236-f009:**
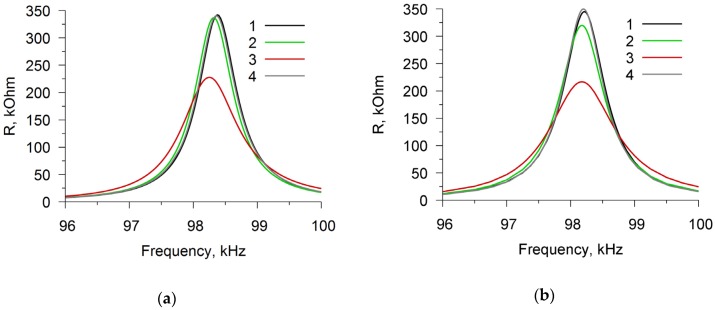
Frequency dependence of the real part of the electric impedance of a resonator with a lateral electric field. The curve 1 relates to the resonator with a 60 μm thick chitosan acetate film in air, the curve 2 relates to the resonator with film in the water vapor, the curve 3 relates to the to the resonator with film in ammonia, the curve 4 relates to the resonator without film in air. (**a**)-experiment, (**b**)–theory.

**Table 1 sensors-20-02236-t001:** The material constants of piezoceramics and chitosan acetate film obtained using acoustic resonance spectroscope of a disk resonator.

Fitted Material Constants of Piezoceramics (Ba_0.24_Pb_0.75_Sr_0.01_(Ti_0.47_Zr_0.53_)O_3_)	
Constant Name	Constant Value
c_11_, 10^10^ Pa	14.96
c_12_, 10^10^ Pa	8.173
c_13_, 10^10^ Pa	8.318
c_33_, 10^10^ Pa	12.84
c_44_, 10^10^ Pa	2.649
e_15_, C/m^2^	15.93
e_31_, C/m^2^	−5.42
e_33_, C/m^2^	17.15
ε_11_	1717
ε _33_	893
ρ, kg/m^3^	7238
**Chitosan acetate film in air mechanical characteristics**	
c_11_, 10^10^ Pa	1.709
c_44_, 10^10^ Pa	1.147
η, 10^−8^ rad/s	8.64
ρ, kg/m^3^	860

**Table 2 sensors-20-02236-t002:** The material constants of piezoceramics and chitosan acetate film used in the calculations of characteristics of a lateral electric field resonator.

Materail Constants of Piezoceramics (Pb_0.95_Sr_0.05_(Ti_0.47_Zr_0.53_)O_3_) (Were Determined Previously by the Authors)	
Constant Name	Constant Value
c_11_, ×10^10^ Pa	10.9
c_12_, ×10^10^ Pa	6.1
c_13_, ×10^10^ Pa	5.4
c_33_, ×10^10^ Pa	9.3
c_44_, ×10^10^ Pa	2.4
e_15_, C/m^2^	10.6
e_31_, C/m^2^	−4.9
e_33_, C/m^2^	14.9
ε_11_	820
ε _33_	840
ρ, kg/m^3^	7330
**Chitosan acetate film in air electrical characteristics**	
ε_11_	1.1
σ, μS/m	56
**Chitosan acetate film in ammonia 1600 ppm electrical characteristics**	
ε	1.1
σ, μS/m	5600
**Chitosan acetate film in water vapor, relative humidity 45% electrical characteristics**	
ε	1.1
σ, μS/m	700

**Table 3 sensors-20-02236-t003:** Data of the ammonia sensors obtained by other authors and in current work.

Sensor Type	Range or Value of Concentration, ppm	LOD, ppm	Sensor Response, %	Sensitivity Coefficient, %/ppm	Response Time– Reset Time	Transducing Mechanism	Reference
Single PPy nanowire	40–300	40	N/A	N/A	15–10 min–15 min	Chemiresistive	[25]
PPy nanowires	1.5–73	1.5	N/A	N/A	60 s for 73 ppm	Chemiresistive	[25]
Single crystal PPy nanotube	0.00005–1	0.00005	N/A	N/A	~16 s–~16 s for 1 ppm	Chemiresistive	[25]
PPy/graphene with TiO_2_ nanoparticles	1–50	1	N/A	N/A	36 s–~16 s for 50 ppm	Chemiresistive	[25]
Graphite oxide	100–500	100	22–30	0.0185	10 min–<10 min	Chemiresistive	[24]
Single-wall carbon nanotubes	62.5–100	N/A	3-6	N/A	N/A	Chemiresistive	[24]
TANIPANI/TiO_2_	20–60	N/A	150–382	5.8	~42 s–150 s	Chemiresistive	[22]
TANIPANI	20–60	N/A	83–256	4.33	~42 s–~172 s	Chemiresistive	[22]
Organic thin-film transistor	0–50	N/A	65	0.227 ± 0.008	N/A–N/A	Chemiresistive	[23]
Chitosan acetate on disk	0–1600	50	105	0.065	N/A–<600 s	Change of the film viscosity	Current work
Chitosan acetate on glass	0–1600	100	98	0.062	N/A–<180 s	Chemiresistive	Current work
Chitosan acetate on resonator with lateral electric field	0–1600	50	−33.2	−0.021	N/A–<300 s	Change of the maximum of the real part of electric impedance	Current work

**Table 4 sensors-20-02236-t004:** Data of the humidity sensors of other authors and obtained in current work.

Sensor Type	Range of Relative Humidity, %	LOD, %	Sensor Response, %	Sensitivity Coefficient, %/%	Response Time–Reset Time	Transducing Mechanism	Reference
Organic film transistor	40–300	40	N/A	N/A	15–10 min–15 min	Chemiresistive	[25]
RGO-PVR film on THS	5–95	N/A	–600%	6.7	2.8 s–3.5 s	Chemiresistive	[3]
Chitosan acetate on disk PZT	20–60	1.5	54	1.35	N/A–<300 s	Change of film viscosity	Current work
Chitosan acetate on glass	20–60	5	96	2.4	N/A–<180 s	Chemiresistive	Current work
Chitosan acetate on resonator with lateral electric field	20–60	5	–4.9	–0.12	N/A–<300 s	Change of the maximum of the real part of electric impedance	Current work

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
