# Peer review of "Evaluation of Elastic Properties and Conductivity of Chitosan Acetate Films in Ammonia and Water Vapors Using Acoustic Resonators [Author-notes fn1-sensors-20-02236]"

_sensors, 2020, doi:10.3390/s20082236_

Round 1
Reviewer 1 Report
In this work, a gas sensor based on piezoelectric resonator with a lateral electric field and a chitosan film was developed. The experiment results showed that chitosan films in response to volatile liquids varied not only conductivity, but also mechanical impedance due to vapor adsorption. Although the work is prepared in a comprehensive manner including the experimental observations, it suffers from fundamental shortcomings that must be addressed carefully before proceeding further.
- What is the effect of vapors on the permittivity? The real-time response and analysis shoulde be given.
- Since the water molecule owns huge permittivity (~80), when the water enters the chitosan film, its permittivity should be changed a lot. So, why the response for ammonia is greater than that for water?
- The sensing mechanism for the coupling between gas adsorption and output performance is missing. Detail interruption and schematic illustration should be given.
- This manuscript investigated the gas sensing performance based on a thin film transducer. Several relative papers may provide the new impact on enriching this work as the references: Nano Energy, 2018, 47, 316-324.; Sensors and Actuators B: Chemical, 2017, 251,144-152; Sensors and Actuators B: Chemical, 2016, 223,202-208; Sensors and Actuators B: Chemical, 2016, 230,330-336;
Author Response
The manuscript: "Gas sensor based on a piezoelectric resonator with a lateral electric field and a gas-sensitive film of chitosan" presents a study in the initial state of a possible application for the determination of gases, but is not proven, in order to publish the authors must follow the following recommendations:
Reply: We thank the Reviewer for this comment. Indeed, this study represents an attempt to evaluate the mechanism of sensing when using chitosan derivatives, like chitosan acetate, as a sensitive layer placed at piezoelectric resonator with a lateral electric field. We show that the changes induced by exposure of such a sensor to gas vapors are mainly determined by both conductivity and mechanical changes of the film (surely this should inlude a mass increase).
To conform to reviewers comment we have change the title of our work and modified the abstract and texts accordingly.
1.pg 1 line 42 the authors indicate as inconvenient that the resistive sensors have a limited life, however, in the proposal they offer they do not study the life of the chitosan film when they put it in contact with ammonia and water vapor.
Reply: We fully agree with the Reviewer. Indeed, when comparing our sensor with chitosan-based resistive sensors, we noted that the latter are less durable in aggressive environments. We meant that metal electrodes are more likely to suffer from such gases than a chitosan film. But since this has not been proved, we remove this phrase from the text and included the possibility of analyzing highly combustible and explosive gases in favor of our sensor. This is due to the fact that the resonator electrodes that are supplied with electrical voltage can easily be isolated from contact with gas. Such a sensor may be useful, for example, in detecting an excessive amount of aviation kerosene vapor in jet aircraft containers. Other things being equal, such a sensor will never give a spark in such a dangerous container.
2.The authors have a previous work published reference 13 where they use chitosan lactate instead of what they propose in this work chitosan acetate, I need to explain better if they have made in this work any modification of the gas chamber with the sensor with respect to the work previously published and that indicate the need to change the reagent.
Reply: As the Reviewer correctly noted, our studies in terms of developing a gas sensor are in the initial stage. Therefore, our work is more methodological in nature. Therefore, as a gas-sensitive film, we tried all the available options for chitosan films. Since in this work we planned to measure the change in the mechanical parameters of chitosan films (longitudinal and shear elastic modules and viscosity coefficient) in the presence of vapor of volatile liquids, then the requirements for the chitosan film, which was deposited on the surface of piezoelectric resonators, were such as plane parallelism of the sides and the absence of roughness of the surface. We were not able to make the necessary films of chitosan lactate and glycolate several tens of microns thick, but these parameters were met with chitosan acetate. This explains the use these films in the work. The flims of chitosan glycolate and lactate appeared to be much thinner. We believe that in the near future we will be able to develop a technology for applying high-quality chitosan films of other types and then we will be able to compare their characteristics.
3.In order to demonstrate efficiency as a sensor for the determination of water vapor and ammonia, it would be necessary to increase the number of experiences by carrying out measurements in a broad range of both gas concentrations, demonstrating that external factors such as temperature and pressure do not affect the results obtained.
Reply: We fully agree with the Reviewer. We measured the elastic modules, viscosity coefficient, and conductivity of chitosan acetate films at different time points with an interval of 45 sec. In this case, the test sample was placed in the chamber and at first the measurements were carried out in air. Then, at time t1, a container with liquid ammonia or water was placed in the chamber, the chamber was tightly closed, and measurements were carried out in the presence of ammonia or water vapor. Then, at time t2, the chamber was opened, and measurements were continued in air. It is obvious that from time t1 to time t2, the concentration of ammonia and water vapor increased, i.e. the film was at different concentrations of the studied vapors. The knowledge of the chamber volume (75 ml), the area of the open surface of the liquid container (2.8 cm2), and temperature (25ºC) allowed us to construct a time dependences of the ammonia concentration and air humidity in the chamber. Thus, in the article we presented the dependences of the elastic modules, the viscosity factor and the conductivity of the film on time and on the concentration of the studied vapors.
As for temperature and pressure, each experiment was carried out twice in laboratory conditions: temperature 26-27ºC, pressure about 1000 hPa, humidity about 20%. In the future, experiments are planned at various temperatures and pressure values.
4.A study of reproducibility and accuracy should be included, as well as stability over time.
Reply: We fully agree with the Reviewer. We have included in “Discussion” the following sentences: “It should be noted that all experiments were carried out at least 2 times and the reproducibility of the results was about ±2%. As for stability over time, the results of our previous work [21], in which the resonator and the chitosan film were separated by an air gap, remain stable within 5% during one year”.

Reviewer 2 Report
The manuscript: " Gas sensor based on a piezoelectric resonator with a lateral electric field and a gas-sensitive film of chitosan" presents a study in the initial state of a possible application for the determination of gases, but is not proven, in order to publish the authors must follow the following recommendations:
1.pg 1 line 42 the authors indicate as inconvenient that the resistive sensors have a limited life, however, in the proposal they offer they do not study the life of the chitosan film when they put it in contact with ammonia and water vapor.
2. The authors have a previous work published reference 13 where they use chitosan lactate instead of what they propose in this work chitosan acetate, I need to explain better if they have made in this work any modification of the gas chamber with the sensor with respect to the work previously published and that indicate the need to change the reagent.
3. In order to demonstrate efficiency as a sensor for the determination of water vapor and ammonia, it would be necessary to increase the number of experiences by carrying out measurements in a broad range of both gas concentrations, demonstrating that external factors such as temperature and pressure do not affect the results obtained.
4. A study of reproducibility and accuracy should be included, as well as stability over time.
Author Response

(The authors gave the same response as above.)

Reviewer 3 Report
This work the authors reported the conductivity of chitosan films changes in vapor of volatile liquids, which has certain sense for the development of sensors. However, there are some issues need to noted:
- Some grammatical or format errors should be corrected.
- I don't found some basic characteristics (such as XRD, SEM, TEM and so on ).
- Why do you choose ammonia as the test gas ? How about other vapor of volatile liquids?
- The vapor concentration is an important parameter in gas sensor field. Thus, the vapor concentration must be controlled
Author Response
Reply: We thank the Reviewer for the positive comments about our work and the constructive criticism.
This work the authors reported the conductivity of chitosan films changes in vapor of volatile liquids, which has certain sense for the development of sensors. However, there are some issues need to noted:
- Some grammatical or format errors should be corrected.
Reply: We tried to correct all errors and inaccuracies. Whenever possible, they are marked with color, since the article has changed significantly, and its volume has increased by about 1.5 times.
- I don't found some basic characteristics (such as XRD, SEM, TEM and so on ).
Reply: We could not employ XRD while chitosan acetate represented a gel, i.e. which is usually characterized by short range order only. Use of TEM is hardly possible for such a delicate material. However, we have performed atomic force microscopy to check the topology of the material which can be considered as an alternative to SEM in this particular case.
So we have included the following text in the Methods:
“Atomic Force Microscopy (AFM) imaging was performed with Bruker Multimode V8 operating in PeakForce tapping regime with HA_CNC cantilevers (k = 1.5 N/m, Optek, Russia).
While drop-casting method was used to prepare the films, we examined the profile of the 10 μl chitosan acetate solution drop-casted on glass substrate after drying in air for 24 h. We employed Dektak150 (Veeco) surface profile measuring system for these measurements.”
We have also included Figure 2 and following text in the Results:
“Study of the film local topology revealed the charachteristic roughnes of the film surface in a dry state of approximately 1 nm (RMS value) depicted in Figure 2a. Still, the dried chitosan droplet casted from 10 μl solution appeared to shows rather smooth profile with ca. 0.5 μm relative deviation (see Figure 2b).”
- Why do you choose ammonia as the test gas ? How about other vapor of volatile liquids?
Reply: Dear reviewer, it is about structure and functional groups of chitosan and its derivatives. The film represents a gel which adsorbes water. The content of water should then depend on the humidity. Moreover, the biopolymer chitosan bears –NH2 functional groups what should make its films successible to ammonia. Moreover, these vapors are polar and should both influence the conductivity and mechanical properties. But the particular influencing mechanism should be different. We tried to address it in the manuscript, in introduction and in the results&discussion section.
The both water and ammonia vapors support change of hydroxyl ion conductivity in chitosan films, i.e. water can protonate free amino groups in the chitosan backbone also forming some hydroxyl ions (-NH2 + H2O → -NH3+ + OH-). This results in the growth of the charge carrier concentration and their mobility which is supported by mechanism of ionic transfer under hydration due to the movement of OH- ions in the electrical field created by NH3+ [Wan, K. A.M. Creber, B. Peppley, V. Tam Bui, Ionic conductivity of chitosan membranes, Polymer 44 (2003) 1057-1065.]. This yields conductivity change, while the adsorption of molecules favors the change in the mechanical properties.
Another reason we chose ammonia and water vapor due to the availability and safety of these substances. In addition, the sensitivity of chitosan films to ammonia and water vapor has not been studied. The only work on the effects of ammonia and water vapor on chitosan is our article [Zaitsev, B.; Fedorov, F.; Semyonov, A.; Teplykh, A.; Borodina, I.; Nasibulin, A. Gas Sensor Based on the Piezoelectric Resonator with Lateral Electric Field and Films of Chitosan Salts. In Proceedings of IEEE Ultrasonics Symp., Glasgow, UK, 6-9 October 2019, pp.607 – 610. DOI: 10.1109/ULTSYM.2019.8925788]. The study raised questions that were partially resolved in this article. But the main goal of the article is to develop a new type of gas sensor based on resonators with a lateral and longitudinal electric field with a gas-sensitive film applied. As a gas sensitive film we used one of the varieties of chitosan - chitosan acetate. The possibility of creating a multi-parameter sensor on this basis was investigated. We planned to use 4 film parameters that change in the presence of gas: longitudinal and shear elastic modules, viscosity coefficient, conductivity and 2 resonator parameters: parallel resonance frequency and the value of the real part of the electrical impedance at this frequency. Thus, it opens the possibility of developing a sensor (analyzer) in which the test gas is characterized by a point in 6 dimensional space. Studies have shown that water vapor and ammonia can be confidently separated by such a combination of parameters. In the future, we will improve research methods and use other gases.
We have included this information in Conclusion.
- The vapor concentration is an important parameter in gas sensor field. Thus, the vapor concentration must be controlled
Reply: We fully agree with the Reviewer. We measured the elastic modules, viscosity coefficient, and conductivity of chitosan acetate films at different time points with an interval of 45 sec. In this case, the test sample was placed in the chamber and at first the measurements were carried out in air. Then, at time t1, a container with liquid ammonia or water was placed in the chamber, the chamber was tightly closed, and measurements were carried out in the presence of ammonia or water vapor. Then, at time t2, the chamber was opened, and measurements were continued in air. It is obvious that from time t1 to time t2, the concentration of ammonia and water vapor increased, i.e. the film was at different concentrations of the studied vapors. The knowledge of the chamber volume (75 ml), the area of the open surface of the liquid container (2.8 cm2), and temperature (25ºC) allowed us to construct a time dependences of the ammonia concentration and air humidity in the chamber. Thus, in the article we presented the dependences of the elastic modules, the viscosity factor and the conductivity of the film on time and on the concentration of the studied vapors.

Reviewer 4 Report
The article is well written, however important experimental work is missing as well as a proper discussion of the results in the text.
The following points should be improved for further considering this work:
Materials and Methods
1) Identify supplier and country. It is not clear to the reader the meaning of “Natural'nyye Ingridienty”.
2) Describe from which reference material parameters were extracted from.
Results
3) Minor error: “equal to”
4) Important details as environment conditions and analyte concentration are missing. It is not clear how many different concentrations and how many times the devices were tested.
5) Avoid reference to colors in the text (both captions and main text). Replace them with different symbols and line styles.
6) Discuss why it takes so long to set the sensor (maybe related to experimental procedure, device structure and film porosity) and why reset can be so fast in some cases.
Discussion
7) Most of the discussion seems like a conclusion. The discussion of the results is actually missing.
8) It is also missing a table comparing these results to organic chemical sensors in literature for the detection of relative humidity and ammonia, as well as a conclusion and more bibliographical references. As an example, the following papers in literature discusses this type of sensors:
Organic sensors for RH and ammonia
Bairi, V.G., Bourdo, S.E., Sacre, N., Nair, D., Berry, B.C., Biris, A.S. and Viswanathan, T., 2015. Ammonia gas sensing behavior of tanninsulfonic acid doped polyaniline-TiO2 composite. Sensors, 15(10), pp.26415-26429.
2. Cavallari, M.R., Izquierdo, J.E., Braga, G.S., Dirani, E.A., Pereira-da-Silva, M.A., Rodríguez, E.F. and Fonseca, F.J., 2015. Enhanced sensitivity of gas sensor based on poly (3-hexylthiophene) thin-film transistors for disease diagnosis and environment monitoring. sensors, 15(4), pp.9592-96093..
3. Bannov, A.G., Prášek, J., Jašek, O. and Zajíčková, L., 2017. Investigation of pristine graphite oxide as room-temperature chemiresistive ammonia gas sensing material. Sensors, 17(2), p.320.
4. Šetka, M., Drbohlavová, J. and Hubálek, J., 2017. Nanostructured polypyrrole-based ammonia and volatile organic compound sensors. Sensors, 17(3), p.562.
Chitosan-based sensors
5. Triyana, K., Sembiring, A., Rianjanu, A., Hidayat, S.N., Riowirawan, R., Julian, T., Kusumaatmaja, A., Santoso, I. and Roto, R., 2018. Chitosan-based quartz crystal microbalance for alcohol sensing. Electronics, 7(9), p.181.
6. Xu, H., Wang, L., Luo, J., Song, Y., Liu, J., Zhang, S. and Cai, X., 2015. Selective Recognition of 5-Hydroxytryptamine and Dopamine on a Multi-Walled Carbon Nanotube-Chitosan Hybrid Film-Modified Microelectrode Array. Sensors, 15(1), pp.1008-1021.
7. Mo, R., Wang, X., Yuan, Q., Yan, X., Su, T., Feng, Y., Lv, L., Zhou, C., Hong, P., Sun, S. and Wang, Z., 2018. Electrochemical determination of nitrite by Au nanoparticle/graphene-chitosan modified electrode. Sensors, 18(7), p.1986.
Conclusions
9) Conclusions are missing.
Considering the points aforementioned, the paper should be rejected.

Author Response
Reply: We thank the Reviewer for the positive comments about our work, the constructive criticism and valuable suggestions.
Comments and Suggestions for Authors
The article is well written, however important experimental work is missing as well as a proper discussion of the results in the text.
The following points should be improved for further considering this work:
Materials and Methods
1) Identify supplier and country. It is not clear to the reader the meaning of “Natural'nyye Ingridienty”.
Reply: We have corrected this error: Chitosan acetate was produced by heterogeneous synthesis, i.e., briefly, chitosan (Bioprogress LCC, Moscow, Russia) with molecular weight, 150-200 kDa.
2) Describe from which reference material parameters were extracted from
We have included the following text in Material and Methods:
“In this study we applied chitosan acetate while it one of the most investigated water-soluble derivative widely available. We plan to measure the change in the mechanical parameters of chitosan films (longitudinal and shear elastic modules and viscosity coefficient) in the presence of vapor of volatile liquids. Therefore the requirements for the chitosan film, which was deposited on the surface of piezoelectric resonator, were such as plane parallelism of the sides and the absence of roughness of the surface. We were not able to make the necessary films of chitosan lactate and glycolate several tens of microns thick, but these parameters were met with chitosan acetate. It also has the easiest solution preparation protocol among the organic acid salts of chitosan (which provide smoother surface, when compared to inorganic acid salts). Chitosan acetate was produced by heterogeneous synthesis, i.e., briefly, chitosan (Bioprogress LCC, Moscow, Russia) with molecular weight, 150-200 kDa, was added to a solution of acetic acid (Sigma Aldrich) in water-ethanol mixture. The mixture was stirred for 3 h at 50 °C. The obtained precipitate was filtered and dried at rotary evaporator using residual pressure 15 mbar and temperature 50 °C. We have prepared a 1.5% aqueous solution using the obtained chitosan acetate. This solution was drop-casted (ca. 0.5 ml) at the electrodes of the piezoceramic resonator or on the glass surface and then was dried in ambient conditions for 24 h. The film’s thickness was measured to vary in the range from 20 to 30 μm. The drop-casting procedure followed by drying was performed several times in the case greater thickness was required”.
Results
3) Minor error: “equal to”
Reply: We have corrected.
4) Important details as environment conditions and analyte concentration are missing. It is not clear how many different concentrations and how many times the devices were tested.
Reply: We fully agree with the Reviewer. We measured the elastic modules, viscosity coefficient, and conductivity of chitosan acetate films at different time points with an interval of 45 sec. In this case, the test sample was placed in the chamber and at first the measurements were carried out in air. Then, at time t1, a container with liquid ammonia or water was placed in the chamber, the chamber was tightly closed, and measurements were carried out in the presence of ammonia or water vapor. Then, at time t2, the chamber was opened, and measurements were continued in air. It is obvious that from time t1 to time t2, the concentration of ammonia and water vapor increased, i.e. the film was at different concentrations of the studied vapors. The knowledge of the chamber volume (75 ml), the area of the open surface of the liquid container (2.8 cm2), and temperature (25°C) allowed us to construct a time dependences of the ammonia concentration and air humidity in the chamber. Thus, in the article we presented the dependences of the elastic modules, the viscosity factor and the conductivity of the film on time and on the concentration of the studied vapors. As for the environment conditions: the temperature is 26-27°C, the pressure is 1000 hPa, the humidity is 20%. We have included this information in the paper.
We have also included in “Discussion” the following sentences: “It should be noted that all experiments were carried out at least 2 times and the reproducibility of the results was about ±2%. As for stability over time, the results of our previous work [21], in which the resonator and the chitosan film were separated by an air gap, remain stable within 5% during one year”.
5) Avoid reference to colors in the text (both captions and main text). Replace them with different symbols and line styles.
Reply: This presentation of the Figures does not contradict the rules of the Journal and therefore we decided not to make changes. But if the Reviewer insists we will correct.
6) Discuss why it takes so long to set the sensor (maybe related to experimental procedure, device structure and film porosity) and why reset can be so fast in some cases.
Reply: The technique of our experiment does not allow us to determine the response time of the film parameter to the presence of vapors with a given concentration. This is explained by the fact that in a closed chamber with an evaporating liquid, the vapor concentration continuously increased. For our chamber, the time for increasing the concentration of ammonia from 0 to saturation (0.17%) is about 2 hours. The time of increasing air humidity from the laboratory value (20%) to saturation (100%) is about 6 hours. As for the different relaxation times for various parameters, further studies are needed to explain.
Discussion
7) Most of the discussion seems like a conclusion. The discussion of the results is actually missing.
Reply: We have included the discussion.
8) It is also missing a table comparing these results to organic chemical sensors in literature for the detection of relative humidity and ammonia, as well as a conclusion and more bibliographical references. As an example, the following papers in literature discusses this type of sensors:
Organic sensors for RH and ammonia
Bairi, V.G., Bourdo, S.E., Sacre, N., Nair, D., Berry, B.C., Biris, A.S. and Viswanathan, T., 2015. Ammonia gas sensing behavior of tanninsulfonic acid doped polyaniline-TiO2 composite. Sensors, 15(10), pp.26415-26429.
Cavallari, M.R., Izquierdo, J.E., Braga, G.S., Dirani, E.A., Pereira-da-Silva, M.A., Rodríguez, E.F. and Fonseca, F.J., 2015. Enhanced sensitivity of gas sensor based on poly (3-hexylthiophene) thin-film transistors for disease diagnosis and environment monitoring. sensors, 15(4), pp.9592-96093..Bannov, A.G., Prášek, J., Jašek, O. and
Zajíčková, L., 2017. Investigation of pristine graphite oxide as room-temperature chemiresistive ammonia gas sensing material. Sensors, 17(2), p.320.
Šetka, M., Drbohlavová, J. and Hubálek, J., 2017. Nanostructured polypyrrole-based ammonia and volatile organic compound sensors. Sensors, 17(3), p.562.
Chitosan-based sensors
Triyana, K., Sembiring, A., Rianjanu, A., Hidayat, S.N., Riowirawan, R., Julian, T., Kusumaatmaja, A., Santoso, I. and Roto, R., 2018. Chitosan-based quartz crystal microbalance for alcohol sensing. Electronics, 7(9), p.181.
Xu, H., Wang, L., Luo, J., Song, Y., Liu, J., Zhang, S. and Cai, X., 2015. Selective Recognition of 5-Hydroxytryptamine and Dopamine on a Multi-Walled Carbon Nanotube-Chitosan Hybrid Film-Modified Microelectrode Array. Sensors, 15(1), pp.1008-1021.
Mo, R., Wang, X., Yuan, Q., Yan, X., Su, T., Feng, Y., Lv, L., Zhou, C., Hong, P., Sun, S. and Wang, Z., 2018. Electrochemical determination of nitrite by Au nanoparticle/graphene-chitosan modified electrode. Sensors, 18(7), p.1986.
Reply: We are grateful to the Reviewer for providing interesting and useful articles. We have included these articles in the introduction and discussion of the paper.
Conclusions
9) Conclusions are missing.
Reply: We have included the conclusions

Round 2
Reviewer 2 Report
I consider that the modified version of the Manuscript "Evaluation of changes of elastic properties and conductivity of chitosan acetate films in gases using acoustic resonators" is suitable for publication in Sensors journal.
Author Response
Dear Reviewer,
Thank you for your comments.
Sincerely,
On behalf of the authors
Boris Zaitsev

Reviewer 3 Report
Authors have carefully addressed the comments and now the manuscript may be published in its current form.
Author Response

(The authors gave the same response as above.)

Reviewer 4 Report
This Reviewer appreciate all the effort performed by the Authors to improve the manuscript.
There are, however, many points that should be addressed and that are clearly present in the attached file.
Title:
It is not clear that the work is on gas sensors.
Abstract:
All the text added after revision need further improvement of the English language. There are many more errors, but a few were highlighted in the abstract and the introduction.
Which concentrations values were tested?
Introduction:
There are phrases that are too long. Split them into shorter ones.
Material and Methods:
Rewrite the phrase describing how data was acquired. How can it be continuous with an interval between measurements.
Please, provide more detail on the calculations of the concentration of analyte in the chamber along time. It can be on a supplementary file.
Results:
For all graphs, avoid any reference to colors (both in captions and text). Use symbols instead. The caption of some figures must be improved too.
What would the sensitivity extracted from these graphs be (i.e. the relative variation of each parameter with respect to a ppm increase, %/ppm)?
What is the limit of detection? Does it saturate? For which values of concentration that happens (if it happens)?
What are the response and reset times of these sensors?
Discussion:
Provide a table comparing these results to the literature (for instance, response time, reset time, sensitivity and limit of detection) that can summarize the discussion.

Author Response
Title:
It is not clear that the work is on gas sensors.
Thank you for this comment. We have corrected the title of the paper.
Abstract:
All the text added after revision need further improvement of the English language. There are many more errors, but a few were highlighted in the abstract and the introduction.
Thank you for your assistance. We have corrected English.
Which concentrations values were tested?
We have introduced the ranges of changing the ammonia concentration (100 – 1600 ppm) and humidity (20 – 60%) in abstract.
Introduction:
There are phrases that are too long. Split them into shorter ones.
Thank for your comment. We have shorted long phrases and corrected English.
Material and Methods:
Rewrite the phrase describing how data was acquired. How can it be continuous with an interval between measurements.
You are right. We measured the frequency dependencies with the interval between measurements of 45 s. In the text we have removed the word “continuously”.
Please, provide more detail on the calculations of the concentration of analyte in the chamber along time. It can be on a supplementary file.
The detailed information about the calculations of the concentration of ammonia and humidity of air is presented in additional file “calculation.doc”
Results:
For all graphs, avoid any reference to colors (both in captions and text). Use symbols instead. The caption of some figures must be improved too.
We have corrected the pointed Figures and figure captions
What would the sensitivity extracted from these graphs be (i.e. the relative variation of each parameter with respect to a ppm increase, %/ppm)?
We have calculated the sensitivity of our sensors and included it in the Tables 1 and 2
What is the limit of detection? Does it saturate? For which values of concentration that happens (if it happens)?
We have calculated LOD of our sensors and included it in the Tables 1 and 2.
What are the response and reset times of these sensors?
Our method of investigation did not allow to estimate the response time. We could only estimate the time relaxation after achievement of the saturation. We have included this information in the Tables1 and 2.
Discussion:
Provide a table comparing these results to the literature (for instance, response time, reset time, sensitivity and limit of detection) that can summarize the discussion.
Thank you for this comment. We have included in the paper the Tables 1 and 2, which allow to make the comparison of some parameters of our sensors with ones of other authors. We have included this comparison in the paper.